# A supervised learning algorithm for interacting topological insulators based on local curvature

Paolo Molignini[1], Antonio Zegarra[2], Evert van Nieuwenburg[3], R. Chitra[4] and Wei Chen[2]

**1** Cavendish Laboratory, University of Cambridge,
19 J J Thomson Avenue, Cambridge, CB3 0HE, United Kingdom
**2** Department of Physics, PUC-Rio, 22451-900 Rio de Janeiro, Brazil
**3** Niels Bohr International Academy, Blegdamsvej 17, 2100 Copenhagen, Denmark
**4** Institute for Theoretical Physics, ETH Zurich, 8093 Zurich, Switzerland

## Abstract

Topological order in solid state systems is often calculated from the integration of an appropriate curvature function over the entire Brillouin zone. At topological phase transitions where the single particle spectral gap closes, the curvature function diverges and changes sign at certain high symmetry points in the Brillouin zone. These generic properties suggest the introduction of a supervised machine learning scheme that uses only the curvature function at the high symmetry points as input data. We apply this scheme to a variety of interacting topological insulators in different dimensions and symmetry classes. We demonstrate that an artificial neural network trained with the noninteracting data can accurately predict all topological phases in the interacting cases with very little numerical effort. Intriguingly, the method uncovers a ubiquitous interaction-induced topological quantum multicriticality in the examples studied.

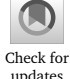
# 1 Introduction

Topological order is typically quantified by an integer-valued topological invariant that is often calculated from the momentum space integration of a certain curvature function, whose precise form depends on the dimension and symmetry class of the system [1–3]. Though the profile of the curvature function in a topological phase varies with the system parameters, the topological invariant remains unchanged. Across topological phase transitions (TPTs) where the topological invariant jumps discretely, the curvature function displays a rather universal feature: [4–7] it gradually diverges at certain high-symmetry points (HSPs) in momentum space, and the divergence changes sign as the system crosses the TPT, causing the discrete jump in the topological invariant. By analyzing the divergence of the curvature function, various statistical aspects of the Landau second-order phase transitions can be transposed to TPTs. These aspects include the notion of critical exponents, scaling laws, universality classes, and correlation functions. These notions form the basis of the curvature renormalization group (CRG) method which can capture the TPTs solely based on the renormalization of the curvature function near the HSP [8], regardless of whether the system is noninteracting [9–13] or interacting [14, 15] or periodically driven [16–19].

The CRG method demonstrates that, although topology is a global property of the entire manifold of the $D$-dimensional Brillouin zone (BZ), the knowledge about topology can be entirely encoded in the curvature function near a HSP. Motivated by this intuition, in this paper we present a supervised machine learning (ML) scheme that utilizes only the curvature function at the HSPs as input data to predict TPTs. The proposed ML scheme answers an important question regarding the application of ML to topological phases: how much data is required to reliably distinguish topological phases? Various ML strategies have been suggested to address this issue, including the concept of quantum loop topography [20, 21], and using either the wave function [22–24], Hamiltonian [25–28], electron density [29], system parameters [30, 31], transfer matrix [32], or density matrix [33] as the input data. Here, we present a simple ML scheme based on input data comprising at most $D + 1$ real numbers in $D$ dimensions applicable to different symmetry classes and weakly interacting systems. We train a simple fully-connected artificial neural network with a single hidden layer with data from prototypical noninteracting TIs whose topological phases are well-known, and then use the trained network to predict the topological phase diagram when many-body interactions are adiabatically turned on such that the single-particle curvature function gradually evolves into its many-body version. We demonstrate how the ML scheme accurately captures the topological phases and phase transitions driven by interaction with very little numerical effort and simultaneously uncover interaction-driven multicritical points. By performing a very simple training procedure (on noninteracting data) we are thus able to access information about the topology in a much wider parameter space, where a direct calculation of the topological invariants is much more cumbersome.

The article is organized in the following manner. In Sec. 2.1, we first review the generic features of the curvature function and the proposed supervised ML scheme based upon it. We then apply this scheme to predict the topological phase diagram of the Su-Schrieffer-Heeger model under the influence of nearest-neighbor interaction in Sec. 2.2 as a concrete example. In Sec. 2.3.1, we apply the ML scheme to 2D Chern insulators with nearest-neighbor interaction, and in section 2.3.2 to Chern insulators with electron-*phonon* interaction, elaborating on the quantum multicriticality caused by the interactions. The results are finally summarized in Sec. 3.

# 2 Machine learning topological phases through local curvature

## 2.1 Supervised machine learning based on local curvature

The topological systems we consider are those whose topological invariant $\mathcal{C}$ is given by a $D$-dimensional momentum space integration

$$\mathcal{C} = \int_{BZ} d^D \mathbf{k} \, F(\mathbf{k}, \mathbf{M}) \,, \tag{1}$$

where $F(\mathbf{k}, M)$ is referred to as the curvature function or local curvature, and $\mathbf{M} = (M_1, M_2 ... M_{D_M})$ is a set of tuning parameters in the Hamiltonian. This form of topological invariant has been proved to be true for any noninteracting system described by Dirac models in any dimension and symmetry class [34]. The points $\mathbf{k}_0$ in momentum space satisfying $\mathbf{k}_0 = -\mathbf{k}_0$ (up to a reciprocal vector) are referred to as the high symmetry points (HSPs). For a $D$-dimensional cubic system, there are $D + 1$ distinguishable HSPs, such as $\mathbf{k}_0 = (0, 0)$, $(\pi, 0)$, and $(\pi, \pi)$ in 2D. Note that $(0, \pi)$ and $(\pi, 0)$ are indistinguishable in the sense that the curvature function has the same value at these two points. As the system approaches the TPT, the $F(\mathbf{k}_0, \mathbf{M})$ generally diverges and flips sign as the system crosses the critical point

$$\lim_{\mathbf{M} \to \mathbf{M}_c^+} F(\mathbf{k}_0, \mathbf{M}) = -\lim_{\mathbf{M} \to \mathbf{M}_c^-} F(\mathbf{k}_0, \mathbf{M}) = \pm\infty. \tag{2}$$

Our aim is to construct a supervised ML scheme to identify the critical point $\mathbf{M}_c$ of TPTs in the $D_M$-dimensional parameter space. Certainly we may use the entire profile of the curvature function $F(\mathbf{k}, \mathbf{M})$ as the input data for ML, but this would be numerically expensive. The question then amounts to what is the minimal amount of data from the curvature function that can accurately predict $\mathbf{M}_c$ with the smallest numerical effort. Since the critical behavior described by Eq. (2) is a defining feature of the TPT, it motivates us to design an ML scheme that uses only the curvature function at the $D + 1$ distinguishable HSPs as input data. Our investigation suggests a supervised ML scheme that consists of the following steps:

(1) We choose a subspace $\widetilde{\mathbf{M}}$ of the parameter space in which all the critical points $\widetilde{\mathbf{M}}_c$ and their corresponding HSPs $\mathbf{k}_0$ at which the curvature function diverges are known.

(2) We generate $F(\mathbf{k}_0, \widetilde{\mathbf{M}})$ for several points in $\widetilde{\mathbf{M}}$, and label them with the value of the corresponding topological invariant. We use this input data to train the neural network for supervised classification of the different topological phases.

(3) Once the neural network is trained, for an unexplored point in the parameter space $\mathbf{M}$, we generate test data $F(\mathbf{k}_0, \mathbf{M})$ at the same HSPs. We then use the trained neural network to predict which phase these points belong to. This last step may be repeated to scan through additional points in the $\mathbf{M}$ space.

Note that interacting subsets may in principle already be used in the training phase. In this work, however, we choose the noninteracting limit as the subspace $\widetilde{\mathbf{M}}$ to train the neural network, whose topology is often easier to solve. We then employ the trained neural network to predict the topology of the parameter space where the interaction is turned on. Our approach assumes that the non-interacting system can indeed manifest nontrivial topology in parts of its parameter space, and that the interacting system is adiabatically connected to the same topological class as the non-interacting one, *i.e.* the interactions do not change the underlying nonspatial symmetries. Because the scheme only relies on the curvature function at the $D + 1$ distinguishable HSPs, it circumvents the tedious integration in Eq. (1) for the interacting cases, and consequently serves as a very efficient tool to obtain the phase diagram in the vast $\mathbf{M}$ parameter space. We now demonstrate the efficiency of our method by studying different interacting TIs.

## 2.2 Su-Schrieffer-Heeger model with nearest-neighbor interaction

We study the 1D Su-Schrieffer-Heeger (SSH) model in the presence of nearest-neighbor interaction to demonstrate the efficiency of the proposed supervised ML scheme. The noninteracting part of the Hamiltonian is given by

$$
\begin{aligned}
\mathcal{H}_0 &= \sum_i (t + \delta t) c_{Ai}^\dagger c_{Bi} + (t - \delta t) c_{Ai+1}^\dagger c_{Bi} + h.c. \\
&= \sum_k Q_k c_{Ak}^\dagger c_{Bk} + Q_k^* c_{Bk}^\dagger c_{Ak} ,
\end{aligned}
\tag{3}
$$

where $c_{Ii}$ is the spinless fermion annihilation operator on sublattice $I = \{A, B\}$ at site $i$, $t + \delta t$ and $t - \delta t$ are the hopping amplitudes on the even and the odd bonds, respectively, and $Q_k = (t + \delta t) + (t - \delta t) e^{-ik}$ after a Fourier transform. We consider the nearest-neighbor interaction [14, 35]

$$
\begin{aligned}
\mathcal{H}_{e-e} &= V \sum_i (n_{Ai} n_{Bi} + n_{Bi} n_{Ai+1}) \\
&= \sum_{kk'q} V_q c_{Ak+q}^\dagger c_{Bk'-q}^\dagger c_{Bk'} c_{Ak} ,
\end{aligned}
\tag{4}
$$

where $n_{Ii} \equiv c_{Ii}^\dagger c_{Ii}$, and $V_q = V(1 + \cos q)$. In the limit of weak interaction, the changes to the topology of the model can be described by renormalizing the Hamiltonian with self-energies calculated from Dyson's equation [14]. To one-loop order, the self-energies are given by

$$
\Sigma_{AA}(k) = \Sigma_{BB}(k) = V ,
\tag{5}
$$

$$
\Sigma_{AB}(k) = \frac{1}{2} \sum_q V_q e^{-i\alpha_{k+q}} = [\Sigma_{BA}(k)]^* ,
\tag{6}
$$

where the phase $\alpha_k$ is defined by $Q_k \equiv |Q_k| e^{-i\alpha_k}$. The $\Sigma_{AA}$ and $\Sigma_{BB}$ are the Hartree terms that introduce a finite chemical potential that shifts the entire spectrum by $-V$. $\Sigma_{AB}$ and $\Sigma_{BA}$ are the Fock terms that modify the off-diagonal elements of the $2 \times 2$ Hamiltonian matrix in the sublattice space. The phase of the modified off-diagonal element then reads

$$
\begin{aligned}
\varphi_k &= -\arg(Q_k + \Sigma_{AB}) \\
&= -\arg\left( Q_k + \frac{1}{2} \sum_q V_q e^{-i\alpha_{k+q}} \right) ,
\end{aligned}
\tag{7}
$$

and the topological invariant is simply the winding number of this phase

$$
\mathcal{C} = \int_0^{2\pi} \frac{dk}{2\pi} \partial_k \varphi_k \equiv \int_0^{2\pi} \frac{dk}{2\pi} F(k, \delta t, V) .
\tag{8}
$$

The curvature function $F(k, \delta t, V)$ with the parameter space $\mathbf{M} = (\delta t, V)$ is thus

$$
F(k, \delta t, V) \equiv \frac{1}{2\pi} \partial_k \varphi_k = \frac{1}{2\pi} \frac{h_x \partial_k h_y - h_y \partial_k h_x}{h_x^2 + h_y^2} ,
\tag{9}
$$

where $h_x \equiv \Re[Q_k + \Sigma_{AB}]$, $h_y \equiv \Im[Q_k + \Sigma_{AB}]$. Note that in the noninteracting limit $V = 0$, the curvature function recovers the more familiar Berry connection [14].

To realize the ML scheme proposed in Sec. 2.1, since the noninteracting SSH model is known to go through TPT via gap closing at $k_0 = \pi$, we use the curvature function $F(\pi, \delta t, 0)$ at $k_0 = \pi$ as the input data to train a neural network that consists of a single dense hidden

layer, as indicated in Fig. 1 (a). The noninteracting $V = 0$ subspace $\widetilde{\mathbf{M}} = (\delta t, 0)$ is used to train the neural network, as indicated by the colored lines in Fig. 1 (c). The details of the training procedure are given in appendix A. In accordance to the usual notation, the $\delta t < 0$ data is labeled as nontrivial with $\mathcal{C} = 1$, and $\delta t > 0$ as trivial with $\mathcal{C} = 0$. After the neural network is trained, we use it to predict the topology in the interacting case $V \neq 0$ in the large $\mathbf{M} = (\delta t, V)$ parameter space. For each $\mathbf{M}$ (darker colored areas in Fig. 1 (c)) we feed the curvature function at the same HSP $F(\pi, \delta t, V)$ to the network to obtain $\mathcal{C}$. The resulting phase diagram shown in Fig. 1 (c) correctly captures the phase boundary between the $\mathcal{C} = 1$ and the $\mathcal{C} = 0$ phases, as can be compared by the results obtained from the curvature renormalization group (CRG) approach [14]. A comparison with Eq. (7) immediately points to the advantage of this ML scheme, because it does not require to an explicit calculation of the highly cumbersome integral in Eq. (8).

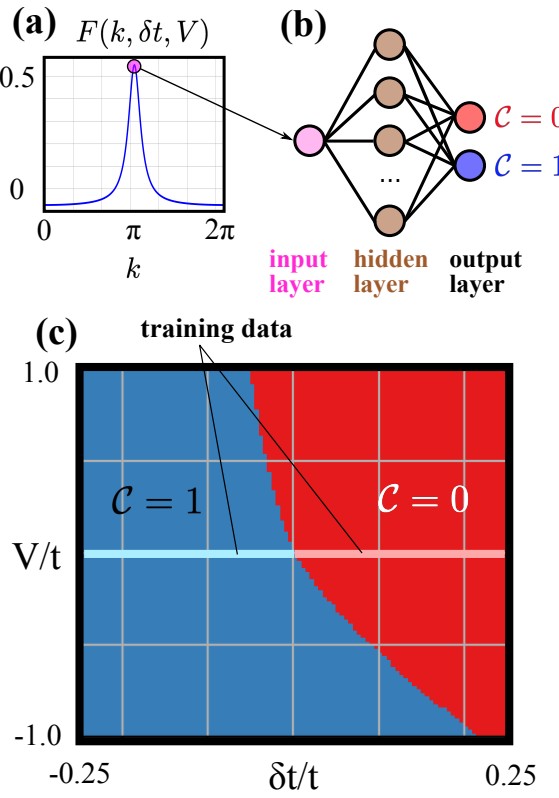

Figure 1: Machine learning scheme to classify different topological phases in the interacting SSH model. (a) The profile of the curvature function $F(k, \delta t, V)$ and the value at the HSP $k_0 = \pi$ used as input data to train a neural network – whose architecture is shown in (b) – to recognize different topological phases. (c) The topological phase diagram predicted by the network for the interacting model ($V \neq 0$), using a one-loop self-energy approximation. The training set is that of the noninteracting SSH model at $V = 0$ given by the topologically trivial phase $\delta t > 0$ (light red line) and the nontrivial phase $\delta t < 0$ (light blue line).

## 2.3 Interacting Chern insulators in 2D

We now apply our algorithm to study interacting TIs in two dimensions. To illustrate the power of the methodology, we consider two kinds of interactions: electronic interactions and electron-phonon interactions. In both cases, we find that the ML scheme predicts a complex

phase diagram and the emergence of interaction-driven multicriticality.

### 2.3.1 Chern insulator with nearest-neighbor electronic interaction

The noninteracting Hamiltonian matrix of the Chern insulator takes the form $H_0 = \mathbf{d}(\mathbf{k}) \cdot \boldsymbol{\sigma}$ in the $(A, B)$ sublattice space for every momentum $\mathbf{k}$, where

$$d_0(\mathbf{k}) = 0, \ d_1 = \sin k_x, \ d_2(\mathbf{k}) = \sin k_y,$$
$$d_3(\mathbf{k}) = M + 2 - \cos k_x - \cos k_y \,. \tag{10}$$

For concreteness, we will examine the nearest-neighbor interaction of a form analogous to Eq. (4), with the vertex

$$V_{\mathbf{q}} = V(2 + \cos q_x + \cos q_y) \,. \tag{11}$$

The effect of the interaction is to modify the Green's function by

$$G^{-1}(\mathbf{k}, i\omega) = \begin{pmatrix} i\omega + d_0' - d_3' & -d_1' + id_2' \\ -d_1' - id_2' & i\omega + d_0' + d_3' \end{pmatrix}, \tag{12}$$

where the $\mathbf{d}$-vector is renormalized by the intra- and inter-sublattice self-energies

$$d_1' = d_1 + \mathrm{Re}\Sigma_{AB} \,, \quad d_2' = d_2 - \mathrm{Im}\Sigma_{AB} \,,$$
$$d_3' = d_3 + \frac{\Sigma_{AA} - \Sigma_{BB}}{2} \,, \quad d_0' = \frac{-\Sigma_{AA} - \Sigma_{BB}}{2} \,, \tag{13}$$

which generally depend on both momentum and energy. The precise form of the self-energies has been discussed previously in detail in [14]. The topological invariant in terms of the full Green's function in this case reads [36, 37]

$$\mathcal{C} = \frac{\pi}{3} \int_{BZ} \frac{d^2\mathbf{k}}{(2\pi)^2} \int_{-\infty}^{\infty} \frac{d\omega}{2\pi}$$
$$\times \epsilon^{abc} \mathrm{Tr}\left[ (G^{-1}\partial_a G)(G^{-1}\partial_b G)(G^{-1}\partial_c G) \right]. \tag{14}$$

Note that $\epsilon^{abc}$ is the Levi-Civita tensor where $\{a, b, c\} = \{\omega, k_x, k_y\}$, and $G \equiv G(\mathbf{k}, i\omega)$ is the interaction-dressed single-particle Green's function. Because the lowest order self-energy is frequency-independent, Eq. (14) greatly simplifies to

$$\mathcal{C} = \frac{1}{4\pi} \int_{BZ} d^2\mathbf{k} \, \hat{\mathbf{d}}' \cdot \left( \partial_{k_x} \hat{\mathbf{d}}' \times \partial_{k_y} \hat{\mathbf{d}}' \right). \tag{15}$$

This form is similar to that of noninteracting 2D class A models, where it simply counts the associated skyrmion number of the self-energy-renormalized $\mathbf{d}'$-vector. The integrand in Eq. (15) is then treated as the curvature function $F(\mathbf{k}, \mathbf{M}) = F(k_x, k_y, M, V)$, with the mass term and interaction strength $\mathbf{M} = (M, V)$ forming a 2D parameter space:

$$F(\mathbf{k}, \mathbf{M}) = \frac{1}{4\pi} \hat{\mathbf{d}}' \cdot \left( \partial_{k_x} \hat{\mathbf{d}}' \times \partial_{k_y} \hat{\mathbf{d}}' \right). \tag{16}$$

We again use a neural network with a single hidden layer to determine the topology in the interacting case, as indicated by Fig. 2 (a), where the curvature function at the three distinguishable HSPs is used as the input data. The noninteracting subspace $\widetilde{\mathbf{M}} = (M, 0)$ is used to train the neural network. The noninteracting subspace has 3 critical points corresponding

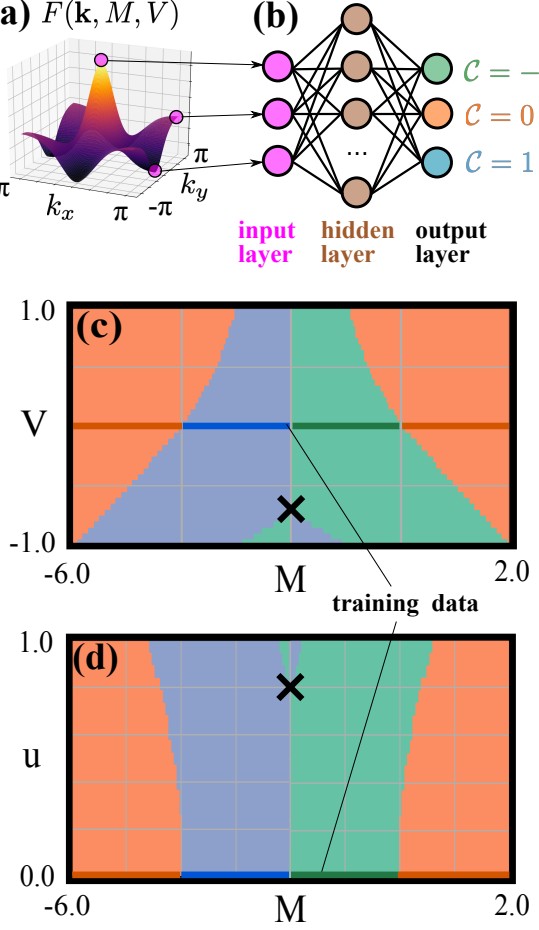

Figure 2: Machine learning scheme to classify different topological phases in interacting 2D Chern insulators. (a) The curvature function at zero frequency $\omega = 0$ and the three inequivalent high-symmetry points $\mathbf{k}_0 = (0,0),(0,\pi),(\pi,\pi)$ used as input data for the neural network – whose architecture is depicted in (b). The ML predicted topological phase diagram for the interacting Chern insulators: (c) electron-electron interactions and (d) electron-phonon interactions. In both cases, the neural network is trained with noninteracting data (shaded lines at $V = 0$ in c) and d)), corresponding to the three inequivalent topological phases with $\mathcal{C} = 0, \pm 1$. The method unveils the existence of multicritical points between the $\mathcal{C} = 1$ and $\mathcal{C} = -1$ phases, indicated by the black crosses.

to the divergence of curvature function at the 3 distinguishable HSPs [38]. Once the neural network is trained, we use it to predict the interacting case $V \neq 0$ in the larger parameter space $\mathbf{M} = (M, V)$, yielding the phase diagram shown in Fig. 2 (c), which correctly captures the three topological phases, as can be compared with the CRG result that has previously solved part of the phase diagram [14]. This again suggests that our ML scheme is a very efficient numerical tool, since it circumvents the cumbersome integration of Eq. (15).

An unexpected result unveiled by our ML method is the prediction of an interaction-driven multicritical point between the $\mathcal{C} = 1$ and $\mathcal{C} = -1$ phases, as indicated by the black cross in Fig. 2 (c) where four regions meet. Although the precise location of this multicritical point and the phase boundaries surrounding it can be altered by higher order self-energy corrections, our result suggests that many-body interactions can be a mechanism for the generation of multicritical TPTs. Such a feature has also been seen in 1D Creutz model with Hubbard-type

interaction [39].

### 2.3.2 Chern insulator with electron-phonon interaction

As electron-phonon interactions are ubiquitous in real materials and can affect properties such as transport of surface states, we now consider the impact of such interactions on the Chern insulator [40–56]. In particular, we consider the deformation potential coupling between an acoustic phonon mode and spinless fermions of the form [57]

$$\mathcal{H}_{e-ph} = \sum_{\mathbf{kq}} M_{\mathbf{q}}(c^\dagger_{A\mathbf{k}+\mathbf{q}} c_{A\mathbf{k}} + c^\dagger_{B\mathbf{k}+\mathbf{q}} c_{B\mathbf{k}})(a_{\mathbf{q}} + a^\dagger_{-\mathbf{q}}), \tag{17}$$

where $a_{\mathbf{q}}$ is the phonon annihilation operator, $\omega_{\mathbf{q}} = v_s q$ is the phonon dispersion with sound velocity $v_s$, and $M_{\mathbf{q}} = u\sqrt{q}$ with $u$ a phenomenological coupling constant determined by sound velocity, electron-ion potential, and ion density. The noninteracting part is that given by Eq. (10). The results for the corresponding self-energies are presented in Appendix B, and extend the calculation of the one-loop self-energies for optical phonons detailed in [14] to the case of acoustic phonons.

We treat the mass term $M$ and the electron-phonon coupling $u$ in Eq. (17) as tuning parameters $\mathbf{M} = (M, u)$, and aim to find the TPTs in this 2D parameter space. A crucial difference from the case of electron-electron interaction in Sec. 2.3.1 is that here, even at the one-loop level, the self-energy depends on both momentum and frequency $\mathbf{K} = (\omega, k_x, k_y)$, and so does the curvature function (integrand of Eq.(14)):

$$F(\mathbf{K}, \mathbf{M}) \equiv \frac{\epsilon^{abc}}{24\pi^2} \mathrm{Tr}\left[(G^{-1}\partial_a G)(G^{-1}\partial_b G)(G^{-1}\partial_c G)\right]. \tag{18}$$

Consequently, the numerical integration of the topological invariant in Eq. (14) becomes even more tedious, especially given the unbounded frequency integration. Nevertheless, we find that close to TPTs, the curvature function at $\omega = 0$ diverges and flips at the HSPs of momentum $\mathbf{k}$. In other words, the appropriate HSPs in this problem are given by $\mathbf{K}_0 = (\omega, k_x, k_y) = (0, 0, 0)$, $(0, \pi, 0)$, and $(0, \pi, \pi)$, at which the critical behavior of $F(\mathbf{K}_0, \mathbf{M})$ follows that discussed in Sec. 2.1. This critical behavior at zero frequency is a reminiscence of gap closures at the Fermi energy at typical quantum critical points that is manifested in the spectral function $A(\mathbf{k}, \omega)$ (detailed in Appendix B).

Our ML scheme becomes a powerful tool in this case, since it circumvents the momentum-frequency integration in Eq. (14), relying instead on the divergence of the curvature function. As in Sec. 2.3.1, we use the noninteracting limit in the absence of phonons $u = 0$ as training data, and apply the ML scheme as illustrated in Fig. 2 (a)-(b). Fig. 2 (d) shows the phase diagram obtained by our ML scheme using the three distinct HSPs $\mathbf{K}_0$ as input data. To check the validity of the results, we plot the spectral function across two representative TPTs (driven by either the mass term $M$ or the electron-phonon coupling $u$), predicted by the ML scheme in Fig. 3. Note that the corresponding spectral functions clearly display a continuous closure and opening of gaps at $\omega = 0$ consistent with a continuous phase transition. This implies that both TPTs driven by the electron-phonon interaction and the mass are second order transitions. To summarize, the phase diagram correctly captures all phases and phase boundaries, and moreover indicates the appearance of a multicritical point as a function of coupling $u$ around $M = -2.0$, indicating that electron-phonon interaction can also serve as a mechanism to induce multicriticality. Thus, many-body interactions are added to the list of several recently uncovered mechanisms that can trigger topological multicriticality, including periodic driving or quantum walk protocols [17–19, 58], long range hopping or pairing [12, 13, 59], spin-orbit coupling [11, 60], topological insulator/topological superconductor hybridization [61], as well as more complicated mechanisms in the spin liquid [62] and toric code models [63].

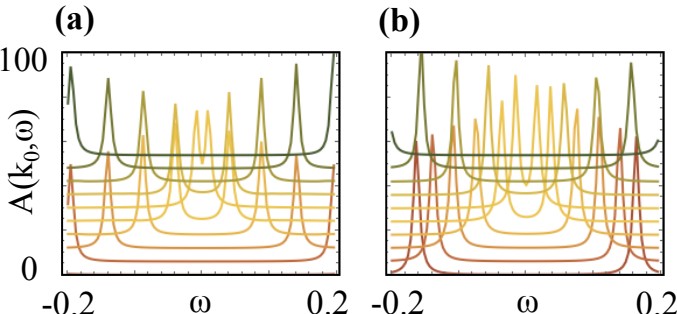

Figure 3: (a) The spectral function $A(\mathbf{k_0} = (0, \pi), \omega)$ for the Chern insulator with electron-phonon interaction. We fix the electron-phonon coupling at $u = 0.4$ and plot $A(\mathbf{k_0} = (0, \pi), \omega)$ for different masses from $M = -2.4$ (bottom curve) to $M = -1.6$ (top curve). One sees a gap-closure at zero frequency at the topological transition point $M_c = -2$. (b) The spectral function $A(\mathbf{k} = (0, 0), \omega)$ at fixed $M = 0.2$ and different couplings from $u = 0.2$ (bottom curve) to $u = 0.8$ (top curve). The gap closure occurs at $u_c \approx 0.5$. These results indicate gap-closures at $\omega = 0$ at the TPTs predicted by the ML scheme, driven by a change in either $M$ or $u$.

We close this section by making a comparison between the CRG [8–15, 17, 18] and the ML scheme proposed here. Though both methods have their advantages and disadvantages, the ML scheme is more efficient than the CRG for obtaining the phase diagram and the related invariants while the latter is more useful to extract critical exponents associated with the TPTs.

## 3 Conclusions

In summary, we propose a supervised machine learning scheme based on the divergence of the curvature function at high-symmetry points, to rapidly identify different topological phases in interacting systems, thereby circumventing costly multi-dimensional integrations. The machine learning scheme consists of an artificial neural network that utilizes as input data $D + 1$ real numbers, representing the values of the curvature function at $D$ distinguishable HSPs in either momentum or momentum-frequency space. The strategy is to train the neural network by the data in a subspace where the topological phases are known – typically the noninteracting case – and then use the trained neural network to predict the topology in a larger parameter space. Because the machine learning scheme circumvents the tedious multidimensional integration of topological invariants, especially in interacting systems, it is a highly efficient tool to map out the topology in a large parameter space regardless the type of interaction and dimension of the system, as demonstrated for several examples. The efficiency of this ML scheme also helps to quickly uncover the multicriticality caused by both the electron-electron and electron-phonon interactions, where multiple topological phases join at a single point on the phase diagram, indicating that these many-body interactions serve as new mechanisms to generate multicritical TPTs. Though the results presented were based on the first order self-energy corrections, a valid approximation for weakly interacting systems, the proposed ML scheme can straightforwardly be extended to higher order self-energy terms. The scheme is widely applicable to topological materials in any dimension and symmetry class, provided the topological invariant is defined from the integration of a local curvature. Future directions include the study of strongly interacting TIs within the paradigm presented here in conjunction with numerical methods like exact diagonalization [15], as well as the interplay of topology and symmetry-broken phases in interacting topological systems.

## Acknowledgements

The authors would like to thank Lode Pollet for useful discussions. This project has received funding from the European Union's Horizon 2020 research and innovation program under the Marie Sklodowksa-Curie grant agreement No. 895439 'ConQuER'. W. C. acknowledges the financial support from the productivity in research fellowship of CNPq. P. M. acknowledges funding from the ESPRC Grant no. EP/P009565/1.

## A  Neural network architecture and training

In this appendix, we give a brief overview of the details of the neural network architecture and training used to obtain the phase diagrams of the interacting topological insulators mentioned in the main text. The construction, training, and evaluation of neural networks was implemented using Tensorflow [64, 65]. For all of the results shown in the main text, we employed a neural network with a single fully-connected hidden layer and varying input and output layer depending on the dimensionality of the system and the number of phases in the topological phase diagram (input: a single neuron for the 1D SSH model, three neurons for the 2D Chern insulators, output: two neurons for the 1D SSH model, three neurons for the 2D Chern insulators). We employed a hidden layer with 10 neurons to generate the results presented in the main text, but we empirically found that the width of the hidden layer can be reduced to 2-3 neurons without significant performance reduction. As activation function, we used a sigmoid for the hidden layer and a softmax for the output layer to obtain classification probabilities. To train the network, we used noninteracting data. We used 4096 points randomly distributed between $\delta t/t = -1.0$ and $\delta t/t = 1.0$ in the SSH model, and between $M = -12.0$ and $M = 12.0$ for the 2D Chern insulator, fed in batches of size 32. The training lasted for 50 epochs. The optimizer used during training was ADAM and the loss function was the categorical cross entropy.

An open-source version of the software used to generate the results presented in this paper is publicly available at https://gitlab.com/paolo.molignini/interacting-topological-insulators-ml.

## B  Self-energy of Chern insulator with electron-phonon interaction

For the Chern insulator with electron-phonon interactions discussed in Sec. 2.3.2, in the zero temperature limit $T \to 0$, taking the Bose distribution $N_0 = 0$ and the Fermi distribution

$n_F(x) = \theta(-x)$, the self-energies are given by

$$
\Sigma_{AA}(\mathbf{k}, i\omega_n) = \sum_{\mathbf{q}} M_{\mathbf{q}}^2
$$
$$
\times \left[ \frac{(1 + d_{3\mathbf{k}-\mathbf{q}}/d_{\mathbf{k}-\mathbf{q}})/2}{i\omega_n - \omega_{\mathbf{q}} - d_{\mathbf{k}-\mathbf{q}}} + \frac{(1 - d_{3\mathbf{k}-\mathbf{q}}/d_{\mathbf{k}-\mathbf{q}})/2}{i\omega_n + \omega_{\mathbf{q}} + d_{\mathbf{k}-\mathbf{q}}} \right],
$$
$$
\Sigma_{BB}(\mathbf{k}, i\omega_n) = \sum_{\mathbf{q}} M_{\mathbf{q}}^2
$$
$$
\times \left[ \frac{(1 - d_{3\mathbf{k}-\mathbf{q}}/d_{\mathbf{k}-\mathbf{q}})/2}{i\omega_n - \omega_{\mathbf{q}} - d_{\mathbf{k}-\mathbf{q}}} + \frac{(1 + d_{3\mathbf{k}-\mathbf{q}}/d_{\mathbf{k}-\mathbf{q}})/2}{i\omega_n + \omega_{\mathbf{q}} + d_{\mathbf{k}-\mathbf{q}}} \right],
$$
$$
\Sigma_{AB}(\mathbf{k}, i\omega_n) = \sum_{\mathbf{q}} M_{\mathbf{q}}^2 \frac{Q_{\mathbf{k}-\mathbf{q}}}{2 d_{\mathbf{k}-\mathbf{q}}}
$$
$$
\times \left[ \frac{1}{i\omega_n - \omega_{\mathbf{q}} - d_{\mathbf{k}-\mathbf{q}}} - \frac{1}{i\omega_n + \omega_{\mathbf{q}} + d_{\mathbf{k}-\mathbf{q}}} \right],
$$
$$
\Sigma_{BA}(\mathbf{k}, i\omega_n) = \sum_{\mathbf{q}} M_{\mathbf{q}}^2 \frac{Q_{\mathbf{k}-\mathbf{q}}^*}{2 d_{\mathbf{k}-\mathbf{q}}}
$$
$$
\times \left[ \frac{1}{i\omega_n - \omega_{\mathbf{q}} - d_{\mathbf{k}-\mathbf{q}}} - \frac{1}{i\omega_n + \omega_{\mathbf{q}} + d_{\mathbf{k}-\mathbf{q}}} \right], \tag{19}
$$

which depend on both momentum and the Matsubara frequency $i\omega_n$. We then replace the Matsubara frequency by a continuous one, $i\omega_n \to i\omega$, in the calculation of the curvature function. The topological invariant is again given by Eq. (14), whose integrand can be expressed in terms of the $\mathbf{d}'$-vector in Eq. (13) by

$$
\frac{\pi}{3} \epsilon^{abc} \mathrm{Tr} \left[ (G^{-1}\partial_a G)(G^{-1}\partial_b G)(G^{-1}\partial_c G) \right]_{\{a,b,c\}=\{\omega,k_x,k_y\}}
$$
$$
= \frac{4\pi i}{\left[ (i\omega + d_0')^2 - d'^2 \right]^2} \left\{ -i\epsilon^{abc} d_a' \partial_x d_b' \partial_y d_c' |_{\{a,b,c\}=\{1,2,3\}} \right.
$$
$$
+ \epsilon^{abcd} d_a' \partial_\omega d_b' \partial_x d_c' \partial_y d_d' |_{\{a,b,c,d\}=\{0,1,2,3\}}
$$
$$
+ i\omega \epsilon^{abc} \partial_\omega d_a' \partial_x d_b' \partial_y d_c' |_{\{a,b,c\}=\{1,2,3\}} \left. \right\},
$$
$$
= F(\mathbf{K}, \mathbf{M}), \tag{20}
$$

where we have denoted $\mathbf{K} = (\omega, k_x, k_y)$. Note that in the noninteracting limit $\mathbf{d}' \to \mathbf{d}$, only the first term in Eq. (20) survives, which recovers the Berry connection $F(\mathbf{k}, \mathbf{M})$ in the integrand of Eq. (15) after a frequency integration. On the other hand, when calculating the spectral function

$$
A(\mathbf{k}, \omega) = -\frac{1}{\pi} \mathrm{Im} \left[ \mathrm{Tr} G^{\mathrm{ret}}(\mathbf{k}, \omega) \right], \tag{21}
$$

we use the retarded version $G^{\mathrm{ret}}(\mathbf{k}, \omega)$ of the interacting Green's function in Eq. (12) obtained through an analytical continuation $i\omega \to \omega + i\eta$.

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
