# Peer review of "A supervised learning algorithm for interacting topological insulators based on local curvature"

_SciPost Physics, doi:SciPost Phys. 11, 073 (2021)_

## Round 1 · Referee Report · Anonymous (Referee 1) · 2021-6-3

Strengths

1- The authors demonstrate how a trained machine learning algorithm can be used to predict the topological invariant of interacting models 2- The authors show that a sparse amount of data allows to successfully predict the topological invariant, turning the method computationally greatly efficient 3- A minimal neural-network architecture is shown to have great efficiency for the task presented

Report

The authors propose an algorithm based on supervised learning to predict the topological invariant of both single-particle and interacting systems. In particular, the authors show that by using a sparse amount of data as a training set for a neural-network, in particular the curvature function at high symmetry points, the trained algorithm is capable of computing the topological invariant of the system. The authors further show that such a trained algorithm can also be applied to many-body data, providing a greatly accurate prediction of the topological invariant.

I found the manuscript of the authors greatly interesting and a very nice demonstration of how a supervised learning algorithm can be used to substantially speed up expensive calculations such as those of interacting topological invariants. I believe that their results are correct and that they are of great interest to the wide communities of topological matter and machine learning. For those reasons, I strongly recommend the publication of their manuscript in Scipost Physics once the minor comment below is addressed.

In their manuscript, the authors focus on models in which the topological phase transition happens by a gap closing at the high symmetry points. While this is often the case, I would like to point out that gap closings can also happen elsewhere in the Brillouin zone. Perhaps a simple example would be a single orbital model in the honeycomb lattice, in which the hoppings in the x-direction are weaker than the others. In this situation, the Dirac points appear somewhere in between the K and M points, and thus small perturbations opening trivial/topological gaps would lead to curvature functions located not in high symmetry points. I think that it could be interesting that the authors comment on whether if their algorithm trained with HSP-data would work for this case, or if one would rather need to retrain it with a more diverse set of k-points. This would, of course, not be an issue at all, yet I think that it could be greatly interesting for the readers to know how large the training set should be for a generic case. The authors do not need to show any new calculations in this regard but rather just comment on what would be the result based on the findings of their manuscript.

To summarize, I believe that their work is of great interest to the readership of Scipost Physics. Therefore, I strongly recommend the publication of their manuscript once the authors briefly comment on the point mentioned above.

Requested changes

1- It could be interesting to mention how important is that the gap closings happen at the high symmetry points for this algorithm to be successful.

---

## Round 1 · Referee Report · Anonymous (Referee 2) · 2021-6-13

Strengths

1 - Novel and computationally efficient way for phase classification of topological insulators (qualifies as an exciting computational discovery) 2 - Uses generalisation neural networks in general provide in a meaningful way: training on computationally accessible data (non-interacting case) and validation and testing on computationally non-trivial data (interacting cases) 3- Method is presented in a clear and understandable way and exemplified on a range a different examples

Weaknesses

1 - The authors advertise their method as the most 'minimal' in terms of input data. I believe this invites potential discussion and criticism: the size of the input their use is small - that is true, but one has to know a lot about a specific model to calculate such inputs. One could easily argue that post-processed experimental data inputs are more 'minimal' as one can calculate, say, small amount of relevant correlations directly from data without knowing anything about the model. The presented method has lot of strengths but the' minimality' as currently formulated is in my opinion questionable and context-dependent.

2 - Authors use rather vague language when describing the training procedure in Section IIA - it's extremely hard to understand how specifically they label data and create the training set (concrete suggestions below)

3 - I did not find a link to OA repository with the code. Given that the results here are purely computational I believe it's appropriate provide the code (ideally the model itself, pre-trained checkpoints and a small data set to test the model on)

Report

In "A supervised learning algorithm for interacting topological insulators based on local curvature" the authors build the phase classification algorithm for topological insulators. The authors use only the value of the curvature function at the high-symmetry points in momentum space and show that this information is sufficient to distinguish the topological phases. Moreover, they show that algorithm trained on easily computable non-interacting data generalises to interacting cases where the curvature function would be expensive to calculate.

I believe the paper meets the Acceptance Expectation 1: computational discovery. Authors formulate novel approach to phase classification with an emphasis of using physics knowledge to compress the training set and the size of the model. Such ventures are particularly meaningful detour from trying to format physics data as input to the traditional image recognition classifiers. Finding minimal representations and effective models yields computational advantage in certain cases (as authors also show) and more readily interpretable models.

Regarding the general acceptance criteria

1- The paper is nicely written, though as pointed out above there are specific points that require further explanation for the sake of clarity 2- Abstract and introduction clearly situate the problem the authors are trying to solve in the context of the field and both are readable and clear (modulo specific overly long abstract sentence I will point out below) 3- Authors do provide sufficient details (modulo specific formulations in the algorithm structure), I believe I would be able to reproduce the results from the description given. However, I would encourage authors to support their verbal descriptions with the code. 4 - The citation list appears to be exhaustive and complete 5 - Again, the description is mostly sufficient (though some hyper parameters like training step size and activations functions are missing) but code would be better 6 - The conclusion, summary and outlook are clear.

Overall, I find this paper will meet the SciPost acceptance criteria if some modification that I list below are made. It is very concisely written and the method presented is a smart solution to complex problem. The presented results provide interesting and relevant insights into the topological phase classification and I believe the results are transferable to other scientists who are building efficient classifiers.

Requested changes

1 - Abstract: The sentence starting "We apply this scheme to a variety.." is too long and hard to follow. Please split in two.

2 - Introduction paragraph 1: In the part starting "Through analysing the divergence... " there are two follow-up sentences starting with "This includes.." and "This forms" - as a reader I lost track what "this" refers to at that point of the text. Please reformulate.

3- Introduction paragraph 2: I have reservation about the "minimal amount of data" statement. As mentioned above, in a different context this means different things and one could argue that minimal amount of data is for example something you can extract from experiment and do not need to calculate based on exhaustive knowledge of the Hamiltonian and wave-function. I would suggest you continue to emphasise that your input has a small size (which is a great merit) but do not claim it is in some sense minimal compared to other approaches ("In contrast to these methods" sentence). I would also suggest that emphasising the simplicity of training and follow-up generalisation are much more important and should be more front and centre in this paragraph as opposed to the input size.

4 - Machine Learning Topological... A, paragraph 2. Please make your 3-point summary of the training process significantly more concrete. "We seek a subspace" - do you seek it in order to create the training set or is that what NN does during training? (I assume the former, but it is not clear from the current statement). "we label them according to" - is the label the value of the topological invariant or is it some function of it ('according to' may mean many things, please be very specific here). I think in the point (3) you talk about how you generate the validation set please make that super specific and don't talk about "asking neural network". How is point (4) different from (3) - is that a continuation of the validation set generation or you are doing something conceptually different. Finally, may the interactions in principle enter already during the points (3) and (4) - if yes please do specify that here since the generalisation is an important point for your paper.

5 - Machine Learning Topological... B, below Eq. 8: since there are no space limitation I suggest you write out F(k,\delta t, V) explicitly. You spent lot of time talking about these functions in the theoretical descriptions of the algorithm, now when you have the concrete example it is helpful to be really explicit about what this function is (it is indeed visible from (7 ) and (8) but it still requires reader to notice it without any help).

6 - Figure 2 - I find the red star notation bit aggressive and not particularly useful for the message of the plot, I imagine that putting something more modest, like diagonal black cross, will have the same expressive result but will be more visually appealing.

7 - Machine Learning Topological... B, pg 5, left column, paragraph 2: again you say $\tilde{F}(K,M)$ is an integrand of large integral from two pages ago - I believe it would be better to write the expression out.

8 - Figure 3: I am bit confused about super thick boundary, it makes the ticks hard to identify. Is there a way to make it thinner? Does it make sense to colour-distinguish the curves or there is a specific reason why all masses have the same colour? I would suggest non-rainbow (for example shades of blue) colouring such that the curves are easier to distinguish.

---

## Round 2 · Author Response

We kindly thank the Referees for their useful comments on our manuscript and their recommendation for publication. In the following, we address all of their comments explicitly.
First Referee
We thank the Referee for their kind words and appreciation of our work. We would like to also address their comment regarding the location of the gap closures. The Referee correctly points out that, in a generic setting, the gap closures might not be located at the high-symmetry points. This could happen in systems where inversion symmetry is explicitly broken, for instance in the graphene-like example suggested by the Referee, or even in a simpler system such as square lattice with a sublattice potential. The gap closures are possibly not pinned at the high-symmetry points anymore, but instead become "mobile", wandering away from the high-symmetry points as the strength of the inversion-symmetry breaking perturbation is increased.
As it is now, our machine learning scheme has been trained and tested only on systems where the gap closures occur at high-symmetry points. However, there are two main strategies that one could adopt to adapt the algorithm to the more general case of gap closures occurring away from the high-symmetry points.
The first, naive solution is to simply map out the gap closures as a function of the tuning parameter and use their shifted position as the data to feed to the machine learning scheme. A straightforward implementation would incorporate the search for the gap closure point which could be numerically cumbersome. The neural network would have to be retrained with this new shifted data.
An alternative approach would be to rely on the conservation of the topological invariant within each phase. Let us assume that the gap closure is shifted away from the high-symmetry point and there is no additional gap closure occurring between this shifted position and the high-symmetry point. This is generically true for the types of two-band Dirac systems analyzed in our work. Then, approaching a topological phase transition, the curvature function will diverge at this shifted point where the gap closure occurs. At the same time, the area under the curvature function, which represents the topological invariant, must remain conserved because of the discrete nature of the topological invariant. The divergence at the gap closure must therefore suck out the weight of the curvature function from its neighborhood to preserve the area under the curve: in other words, all points in its vicinity will experience a continuous, monotonic change in the value of the curvature function to compensate for the divergence. This change will have a direct correspondence with the value of the topological invariant, and therefore can also be taken as the input data to characterize the topology. In particular, these points can be taken to be high-symmetry points. We remark that this scenario does indeed appear for topological phase transitions associated with frozen dynamics in periodically driven systems [see for instance Phys. Rev. B 102, 235143 (2020) and Phys. Rev. B 98, 125129 (2018)]. In this context, the gap closures occur at non-high-symmetry points, but by mapping out the behavior of the curvature function at high-symmetry points one can still determine the correct phase diagram. Because the divergent behavior is consistent across all topological phase transitions (of the same universality), the high-symmetry points can then still be fed to the machine learning scheme as input data. The neural network should automatically learn to associate the change in value at the high-symmetry points with the given topological phase, even though the gap closures occur elsewhere.
Second Referee
We deeply appreciate the Referee’s positive reception of our work and thank them for their recommendation to publish the manuscript. We have fixed all the issues raised by the Referee. In particular: - Following the Referee’s suggestions, we have reformulated several sentences in the abstract and introduction to improve the clarity of our message. - We have reformulated our statement about the “minimal amount of data” required to predict the topology of a given phase. We have also highlighted the simplicity of our training and its generalization rather than the input size. - We have substantially reworked our summary of the training and testing procedure to improve its clarity (note also the reduction from 4 to 3 points). - As requested by the Referee, we have written the explicit formulas for the curvature functions. Note that for the Chern insulator, expanding out the formulas would generate expressions that would occupy an entire page. - We have slightly reworked Figs. 2 and 3 to accommodate the Referee’s requests. - We have also included a link to an open-source repository of the python code that was used to generate the results of the paper.

---

## Round 2 · List of Changes

- Slight changes to the abstract and introduction.
- Added more explicit formulas for the curvature functions.
- Slightly changed the appearance of Figs. 2 and 3 (no change in the data).
- Reformulated the summary of the machine learning procedure. It now consists of three main steps and not four.
- Added link to open-source repository.

---

## Editorial Decision

published